# Performance Evaluation of Zone-Based In-Vehicle Network Architecture for Autonomous Vehicles

**DOI:** 10.3390/s23020669

**Published:** 2023-01-06

**Authors:** Chulsun Park, Sungkwon Park

**Affiliations:** Department of Electronic and Computer Engineering, Hanyang University, Seoul 04763, Republic of Korea

**Keywords:** in-vehicle network (IVN), zone-based IVN architecture, domain-based IVN architecture, wiring harness, length, weight, end-to-end delay

## Abstract

In recent years, various functions such as advanced driver assistance systems (ADAS) and infotainment systems are being mounted in vehicles for safety and convenience to drivers. Among the various functions, autonomous driving-related technologies are being added to all vehicles, from low options to high options. For autonomous driving, hundreds of new electronic control units (ECUs) including various advanced sensors would be needed. Adding more ECUs would enhance safety and convenience for the driver. On the other hand, wiring between these ECUs would be more complex and heavier. The wiring harness is essential for communication and power supply. Currently, the in-vehicle network (IVN) uses the domain-based IVN architecture (DIA) that separates ECUs into domains based on their functions. Recently, in order to minimize the complexity of wiring harness and IVN, zone-based IVN architecture (ZIA) that groups ECUs according to their physical locations is attracting attention. In this paper, we propose a new DIA and ZIA for autonomous driving in the context of time-sensitive networking (TSN). These two new IVN architectures are simulated using the OMNeT++ network simulator. In the simulation process, a mid-size vehicle is assumed. It is shown in this paper that ZIA not only reduces wiring harnesses in both lengths and weights by approximately 24.6% compared to the DIAs, but also reduces data transmission delay.

## 1. Introduction

As the competition to develop autonomous vehicles intensifies around the world, technologies related to autonomous driving are attracting attention. The autonomous vehicles can provide convenience and safety to drivers and pedestrians by providing services such as reducing traffic accidents, improving driving accessibility, and allowing various non-driving tasks while moving. In recent years, the original equipment manufacturers (OEMs) and information technology (IT) companies are making huge investments in the development of autonomous vehicles, and the autonomous driving technologies and these markets are growing rapidly [1]. According to the document [2] published by Precedence Research, the global autonomous vehicle market is estimated to grow from $94.4 billion in 2021 to about 1808.4 billion in 2030. In addition, the compound average growth rate (CARG) is expected to grow to 38.8% from 2021 to 2030. The society of automotive engineers (SAE) divided the steps of autonomous vehicles into six levels depending on the presence or absence of driver intervention [3]. Most of the autonomous vehicles currently under development are in level 3. Many global OEMs and IT companies are aiming to develop level 4 or higher autonomous vehicles within a few years.

As the level of autonomous driving increases, the number of electronic control units, (ECUs) including sensors such as cameras (CAMs), radar, and LiDar mounted in vehicles that control them, would also increase significantly compared to that in the existing vehicles. Approximately 70 ECUs must be mounted in a vehicle to provide advanced driver assistance systems (ADAS) functions [4]. The level of autonomous driving can be improved by equipping more ECUs in a vehicle. However, this increases the complexities of the in-vehicle network (IVN) architectures. Since communication between ECUs is achieved through wiring harnesses, an increase in both the lengths and weights of the wiring harnesses is inevitable. This increase worsens the complexity of IVN architectures, which not only increases vehicle weight and manufacturing cost but also negatively impacts data transmission delays and other quality of service (QoS) parameters. An efficient IVN architecture is always needed to solve these problems. Since the wiring harnesses are the third heaviest after the engine and chassis, it takes up a significant portion of the vehicle [5].

To develop fully autonomous vehicles, various studies are needed. This paper focuses on IVN protocol and IVN architecture among various studies. To transmit time-sensitive traffic generated by autonomous vehicle ECUs, an IVN protocol that meets the requirements of wide bandwidth, low delay, and lossless is required. Time-sensitive traffic includes audio and video data and control data that can affect driver and vehicle safety, such as airbags, brakes, and driver control. Traditional IVN protocols include local interconnect network (LIN), controller area network (CAN), and media-oriented system transport (MOST). However, as ADAS and infotainment systems are adopted in autonomous vehicles, communication requirements on IVN such as wide bandwidth support, low delay, and loss for time-sensitive data are increasing. It is becoming more difficult for the traditional IVN protocols to meet these requirements. Recently, research on Ethernet-based IVN protocols is becoming more active. These include IEEE 802.1 Ethernet audio video bridging (AVB) Gen 1 and Gen 2. They have been all developed in the context of the time-sensitive networking (TSN). These Ethernets can support a wide bandwidth of 100 Mbps to 10 Gbps and low data transmission delay. In addition, since they have high compatibility with various heterogeneous devices based on Internet protocols (IP), various IP services can be easily provided through vehicle-to-everything (V2X). IEEE 802.1 Ethernet AVB is being developed to achieve various levels of the time sensitivity for transporting audio and video data. This has been extended to IVN in the last decade. The IEEE 802.1 TSN Task Group (TG) has been standardizing TSN technologies since 2012 [6]. TSN consists of various standards such as timing synchronization between nodes, forwarding, queuing, and bandwidth allocation.

Even if the IVN of an autonomous vehicle supports these TSN protocols, it may be difficult to transmit time-sensitive traffic with an improperly designed IVN architecture properly. As previously mentioned, the IVN architecture plays important role in the lengths of the wiring harnesses and its weights. Until 2018, vehicles used a distributed IVN architecture. This architecture consists of connecting multiple ECUs to one gateway. As a result, a large amount of data is transmitted through the central gateway, causing a serious bottleneck. In addition, it is difficult to add new ECUs to a vehicle due to the limitation of the scalability and flexibility of the central gateway. The domain-based IVN architecture (DIA) was developed to overcome the difficulty. DIA is now widely used in vehicles. DIA has a structure in which ECUs are grouped into several domains according to their functions. DIA can alleviate bottlenecks introduced by previous architecture. In addition, since ECUs with similar functions are grouped, DIA has the advantage of being able to process complex functions quickly compared to distributed IVN architecture [7]. However, DIA requires a long wiring connection because it connects to one controller without considering the location of the ECUs. This results in the increased complexity on the vehicle wiring and network architecture. According to [8], the weight of the wiring harnesses used for DIA is up to 80 kg, and the length is up to 5 m. As more ECUs are mounted in fully autonomous vehicles, the complexity becomes even more exacerbated. To solve these problems of DIA, zone-based IVN architecture (ZIA), which groups ECUs according to their physical location, has recently been attracting attention. According to [9,10], ZIA expects to reduce vehicle weight and manufacturing costs by reducing the weights and lengths of the wiring harnesses compared to DIA. As an additional advantage, this architecture is expected to enable fast data processing and transmission if this architecture is adopted for vehicles. However, there are no specific studies that numerically compare DIA and ZIA in terms of data transmission and the lengths and weights of the wiring harnesses. Furthermore, design studies for adopting ZIA in autonomous driving or future vehicles are in their early stages.

This paper verifies through simulation whether ZIA, which has recently been attracting attention, is suitable as an architecture for autonomous vehicles and how superior it is to DIA. There are various evaluation elements to verify excellence, but this paper evaluates three elements. The three evaluation elements are end-to-end (E2E) delay and the total length and weight of the wiring harnesses. In TSN, the primary evaluation element is E2E delay. The advantages of ZIA are to reduce the lengths and weights of wiring harnesses and simplify connections. For these reasons, this paper evaluates both architectures by focusing on three elements aspects. We design new DIA and ZIA based on TSN for autonomous driving by referring to technical documents and papers related to the internal network of autonomous vehicles, then use the OMNeT++ network simulator to develop and simulate simulators for DIA and ZIA. The results obtained through simulation are analyzed to verify the excellence of ZIA and its suitability as an autonomous vehicle architecture.

The structure of this paper is as follows. Section 2 introduces the IVN protocols based on automotive Ethernet and the evolution of the IVN architectures. Section 3 describes how to design DIA and ZIA architectures. Section 4 describes the development of simulators for the two IVN architectures using the OMNeT++. Section 5 describes the experimental results on the E2E delays of data transmission and the lengths and weights of the wiring harnesses derived through simulation. Finally, Section 6 describes the conclusions and future works.

## 2. Background and Related Works

This section describes studies related to background knowledge. Sub-section A describes IEEE 802.1 Ethernet AVB (Ethernet AVB) Gen 1 and Gen 2. Sub-section B describes the IVN architectures used in the past and to be potentially used in the future. Additionally, we would like to introduce the ZIA in more detail.

### 2.1. IEEE 802.1 Ethernet AVB Gen 1 and Gen 2

The IVN architectures in autonomous vehicles should satisfy very stringent communication requirements. This is because ECUs for autonomous driving must transmit and receive time-sensitive data. For example, radars and LiDars send data, and the brain function ECU must decide to stop or go within a limited time to prevent accidents. The traditional IVN protocols before Ethernet AVB have difficulties in transmitting and receiving time-sensitive data. More ECUs are being mounted in vehicles to enhance the levels of autonomous driving. Traditional IVN protocols such as LIN, CAN, and MOST provide low-speed communication, making it difficult to send and receive time-sensitive data generated by ECUs within a limited time. In order to solve this difficult, research is being conducted to adopt the Ethernet AVB standards to time-sensitive automotive Ethernet.

Ethernet AVB was developed to transmit data with accurate time synchronization between devices that transmit time-sensitive audio and video (AV) data especially in a broadcasting company. Without a certain level of time sensitivity, AV data may not be synchronized each other. This may annoy viewer or listeners. Ethernet AVB consists of three standards: IEEE 802.1 AS, Qat, and Qav.

Synchronization between all devices on a network. The time synchronization process specified in the IEEE 802.1 AS standard is as follows. Firstly, one of the devices is selected as a grand master (GM), and the selected GM provides a reference time to all devices on the network. The GM is selected using the best master clock algorithm (BMCA). After GM is selected on the network, the spanning tree protocol is used to generate an optimal path for data transmission. Finally, the time of the distributed devices along the path is synchronized to the time of the GM. In order to transmit the AV data, the time synchronization of devices on the network is essential.

The IEEE 802.1 Qat defines the types of traffic transmitted in networks using Ethernet AVB and specifies how bandwidth is allocated to ensure their temporal requirements. The types of traffic used in Ethernet AVB are classified into three classes: stream reservation (SR) Class A traffic, B traffic, and best effort (BE). The SR Class A traffic and B traffic are classes for audio and video data transmission. Among the classes, the SR Class A traffic has the highest priority, and the BE has the lowest priority for transmission. The IEEE 802.1 Qat specifies the maximum E2E delay in data transmission and various requirements for each class. The E2E delay requirement for the SR Class A traffic is within 2 ms over 7 hops and for the SR Class B traffic it is within 50 ms over 7 hops [11]. Finally, the E2E delay requirements for BE are not specified in the standard. Additionally, the IEEE 802.1 Qat defines stream reservation protocol (SRP) for network resource reservation to ensure AV data transmission quality.

IEEE 802.1 Qav defines forwarding and queuing for preferential transmission of AV data inside a switch. When the three types of traffic defined in IEEE 802.1 Qat are transmitted, the switch forwards them to the output port according to the priority of each traffic. The switch uses two algorithms defined in the IEEE 802.1 Qav: credit-based shaper (CBS) and strict priority (SP). The CBS is used for the SR Class A and B, and the SP is used for BE.

Ethernet AVB TG has developed Ethernet AVB Gen 1 for algorithms and protocols to ensure low-delay data transmission and synchronize AV data. Ethernet AVB Gen 1 was a standard for the transmission of AV data related to infotainment systems. On the other hand, time-sensitive control data used in ADAS must have more stringent communication requirements the AV data. To meet these requirements, IEEE 802.1 TSN TG is developing Ethernet AVB Gen 2. The TG is developing various standards for deterministic data transmission, low latency, and lossless and reliable data transmission on time-sensitive automotive Ethernet. New scheduled traffic (ST) was added to Ethernet AVB Gen 2 for time-sensitive control data transmission. The ST has shorter E2E delay requirements than other types of data. The ST must be transmitted within 100 us when going through 5 hops [12]. Ethernet AVB Gen 2 consists of various standards. IEEE 802.1 Qbv defines a time-aware shaper (TAS) algorithm for transferring time-sensitive data from a switch before other data. IEEE 802.1 Qbu defines preemption and guard bands (GB) to ensure time-sensitive data transmission. Preemption is a technology that stops data transmission and preferentially transmits time-sensitive data. IEEE 802.1 Qcc defines techniques for improving SRP and performance. IEEE 802.1 Qca defines the route control and reservation method. In addition to these, Ethernet AVB Gen 2 includes several additional standards. These standards can be found in the IEEE 802.1 TSN TG.

### 2.2. Evolution of IVN Architectures

Until 2019, vehicles used distributed IVN architecture with all ECUs connected to a central gateway [13,14]. This architecture is suitable as an IVN architecture for vehicles with a small number of ECUs and transferring and receiving small amounts of data. However, with the development of autonomous driving technologies, the number of ECUs mounted in vehicles is increasing more than in the past. In this architecture, communication takes place between all ECUs through a central gateway, which inevitably places a higher load on the gateway compared to other architectures. A high load on the gateway causes data losses and delays. Data losses and delays in autonomous vehicles can lead to serious traffic accidents. This architecture can increase the complexity of wiring and network as all ECUs are connected to a central gateway. As complexity increases, networking performance and time-sensitive communication speeds decrease.

Currently, OEMs are adopting DIA for their vehicles. Figure 1 is an example of DIA that groups ECUs into five categories: ADAS, powertrain (PT), chassis, body, and infotainment. Looking at the structure of DIA, DIA consists of one gateway and multiple domain controllers. Because DIA groups ECUs with similar functions, data transfer is primarily within a domain. Since data is rarely transmitted outside the domain, this architecture can reduce the load on a central gateway compared to the previous architecture. However, since DIA connects to one domain controller without considering the location of the ECU, as the number of ECUs increases, the lengths and weights of the wiring harnesses are adversely affected. For example, cameras or sensors are placed on the edge of the vehicle and require long wiring harnesses to connect to the domain controller. As a result, the increase in the lengths and weights of the wiring harnesses not only increases the weight and manufacturing cost of the vehicle but also adversely affects data transmission delay. As various technologies, such as V2X and over-the-air (OTA), are added to future vehicles and develop into fully autonomous vehicles, more ECUs will be required in vehicles. Therefore, DIA is unsuitable as an IVN architecture for future or autonomous vehicles. To overcome the problems of DIA, OEMs and auto parts manufacturers are proposing ZIA as the next-generation IVN architecture.

ZIA is an architecture that groups ECUs into zones based on physical proximity rather than grouping them according to similar functions. Figure 2 shows a ZIA that groups ECUs into six zones according to their physical location. As shown in the figure, one zone controller is placed in each zone, and a high-performance computing unit is placed in the center of the ZIA. DIA is a domain-centric IVN architecture, and ZIA is a centralized IVN architecture. In DIA, multiple domain controllers perform data collection, calculation, and control according to their functions, but in ZIA, a single high-performance computing unit performs all tasks. This unit works like a brain and also acts as a central gateway to pass data from one zone to another [13]. Autonomous vehicles require high-performance autonomous applications that collect data from multiple sensors, make situational decisions, and control the vehicle. Therefore, ZIA, which can perform collection, calculation, and control functions through a single high-performance computing unit, is suitable as an architecture for autonomous vehicles. The most significant advantage of adopting ZIA to vehicles is the simplification of communication networks and wiring harnesses [14]. As a result, ZIA can benefit from faster data processing and transmission than previous IVN architectures.

## 3. Design of IVN Architectures for Autonomous Vehicles

This section describes how to design DIA and ZIA for autonomous vehicles based on automotive Ethernet (Ethernet AVB Gen 2) before simulation. According to [15], today’s SAE Level 1 and 2 luxury vehicles are equipped with around 150 ECUs. Among the 150 ECUs, there are also ECUs that are less relevant to time-sensitive traffic, such as doors, trunks, and windows. These ECUs have nothing to do with autonomous driving. Although we can design DIA and ZIA with 150 ECUs, the primary purpose of this paper is to compare data transmission E2E delays and the lengths and weights of the wiring harnesses according to the structural characteristics of IVN architectures for autonomous vehicles. Therefore, in this paper, we design new DIA and ZIA for autonomous vehicles by selecting 36 ECUs closely related to autonomous driving.

Suppose we can obtain detailed information from OEMs, such as the locations of the ECUs and the lengths and weights of the wiring harnesses. In this case, we can design DIA and ZIA similar to the actual IVN architectures. However, the security policy makes it difficult to obtain this detailed information. We refer to IVN structural dimensions data provided by OEM and [16,17,18,19,20,21,22] to design DIA and ZIA proposed in this section. The wiring harnesses have different numbers of sockets, connectors, and wires, depending on their function. In addition, when connecting devices, the wiring harnesses are connected in a complex curve rather than a straight line. In this paper, DIA and ZIA are designed assuming that the ECUs and controllers are connected in a straight line with the same type of wiring harness.

### 3.1. Domain-Based IVN Architecture

Figure 3 shows DIA with automotive Ethernet designed in this paper. DIA consists of ADAS, PT and chassis, infotainment, and body. The ADAS domain contains ECUs that are closely related to autonomous driving. This domain includes sensors such as radar, LiDar, global positioning systems (GPS), ultrasonic, and various CAMs. ADAS sensor fusion and detection (FD) collects and analyzes data from the ECUs including sensors to generate data necessary for autonomous driving. Radar and LiDar identify distance and road conditions from surrounding objects. GPS is used to precise the position of the vehicle using satellites. Ultrasonic sensors are used to measure the distance and speed between vehicles. The various CAMs in different parts of the vehicle are used to identify the surroundings and driver conditions. The various CAMs include mirrorless, around view monitoring (AVM), driver state monitoring, front, infrared, and rear CAMs. Other domains other than the ADAS domain include ECUs such as head up display (HUD), antenna, monitors, electronic stability controls (ESCs), motor driven power-steering (MDPS), audio mixing platform (AMP), etc.

DIA designed in this paper consists of a domain gateway and four domain controllers. A 1 Gbps wired links were used to connect DIA backbone network and the ADAS domain controller and the ECUs. The red lines are the 1 Gbps wired links. Connections between other domain controllers and ECUs used the 100 Mbps wired link. The blue lines are the 100 Mbps wired links. The ECUs, which are closely related to autonomous driving, use the 1 Gbps wired links because they not only generate a large amount of time-sensitivity data but also transfer and receive data within a limited time. For example, CAMs generate more AV data than other ECUs. High-bandwidth links are required as road information collected via CAMs need to be quickly identified and time-sensitivity data transferred.

### 3.2. Zone-Based IVN Architecture

OEMs and vehicle component manufacturers present ZIAs that group IVN from a minimum of four to a maximum of 13 zones. Figure 4 is ZIA with automotive Ethernet that we designed. We design ZIA, which groups IVN into six zones. There are six zones: front left, front right, center, rear, rear left, and rear right. The number of ECUs included in ZIA equals the number of ECUs containing DIA designed in Figure 3. ZIA presented in this paper consists of six zone controllers. ZIA backbone network and autonomous driving-related ECUs were connected to nearby zone controllers using the 1 Gbps wired links. Other ECUs were connected to zone controllers using the 100 Mbps wired links. The red lines are the 1 Gbps wired links and the blue lines are the 100 Mbps wired links.

When designing ZIA, various network topologies such as ring, star, tree, and mesh can be adopted. There is no right answer as to which topology to use when designing network architectures. Developers can use the appropriate topology considering network conditions. In this paper, ZIA is designed based on a star topology that is advantageous in data transmission. In future experiments, we may also consider tree, ring, and mesh rather than a star.

## 4. Development of DIA and ZIA Simulators

This section describes the development of a simulator to compare the E2E delays and the lengths and weights of the wiring harnesses according to the structural characteristics of the IVN architectures. Using the OMNeT++, we develop the simulators for DIA and ZIA designed in the previous section. Through the developed simulator, we verify how superior ZIA is to DIA in terms of data transmission and the lengths and weights of the wiring harnesses.

### 4.1. Simulation Environment

We develop DIA and ZIA simulators using the OMNeT++ (Ver.5.6.2) [23] and the open-source CoRE4INET framework [24]. The CoRE4INET framework provides functions conforming to the Ethernet AVB Gen 1 and Gen 2 standards using the INET framework. Figure 5 shows DIA developed using the OMNeT++ to simulate a previously designed DIA. This DIA is an IVN architecture grouped into four domains according to functions. Figure 6 shows ZIA developed using the OMNeT++ to simulate a previously designed ZIA. This ZIA adopts a star topology and is an IVN architecture grouped into six zones according to physical locations of ECUs.

### 4.2. Simulation Method

Using the previously developed simulators, we compare the E2E delays for time-sensitive data transmitted both from DIA and ZIA. Furthermore, the lengths and weights of the wiring harnesses required to design DIA and ZIA proposed in this paper are compared. First of all, we define the different traffic types generated by ECUs to compare the E2E delays for data transmission occurring in the two IVN architectures. In order to define the traffic types, we refer to traffic-related information provided by OEMs and another reference [25]. We developed simulators that generate a total of 15 traffic from the ECUs of the DIA and ZIA designed above. Table 1 defines the types of traffic generated by ECUs deployed on DIA and ZIA. This table includes traffic priorities, Ethernet payload sizes, traffic intervals, and transmission paths according to the traffic types. The traffic types are classified into the ST, SR Class A, and B according to data transmission priorities. The ST is data necessary for autonomous driving such as control, V2X, LiDar, navigation, and PT and chassis data. The SR Class A and B include AV and infotainment data, etc. Ethernet AVB Gen 1 and Gen 2 standards define the minimum time intervals between traffic. These standards define BE in addition to the ST, SR Class A, and B. However, BE has the lowest priority of the four traffic and is not used to transfer time-sensitive traffic. Therefore, BE was excluded from this paper.

We measure the E2E delay for 15 traffic types generated by the ECUs defined in Table 1. In the process of data transmission, it is best to transmit data without delay. However, in actual network environments, delays occur due to various factors. There are four types of delays that occur in the process of data transmission on a network. The four types are processing, queuing, transmission, and propagation delays. The processing delay occurs inside switches. This delay includes the time it takes to determine the output port for data transfer inside the switch and to check for data errors. This delay depends on the performance of the processor. The queuing delay is the time during which data wait in the queue before being delivered to the output port. This delay is affected by congestion inside the switch. The transmission delay is the time it takes to deliver data from the output of the queue to the transmission medium. In DIA and ZIA proposed in this paper, types of transmission media are the wired links with 1 Gbps and 100 Mbps bandwidths. The transmission delay depends on the bandwidth of the links. Finally, the propagation delay is the time it takes for the data to be transmitted to the destination through the link. The E2E delay of data generated by ECUs in DIA and ZIA can be expressed by Equation (1).
(1)$$E2E\;Delay=\overset{N_{L}}{\underset{i=1}{\sum }}L_{Prop\_Delay_{i}}+\overset{N_{S}}{\underset{j=1}{\sum }}S_{Delay_{j}}$$

$L_{Prop\_Delay_{i}}$ is the propagation delay caused by the $i^{th}$ link. $S_{Delay_{j}}$ are delays that occur inside the $j^{th}$ switch. $N_{L}$ is the number of links, and $N_{S}$ is the number of switches.

Equation (2) shows the sum of the delays occurring inside the switch.
(2)$$S_{Delay}=S_{Proc\_Delay}+S_{Que\_Delay}+S_{Trans\_Delay}$$

The $S_{Proc\_Delay}$ is the processing delay and $S_{Que\_Delay}$ is the queuing delay. Finally, $S_{Trans\_Delay}$ is the transmission delay.

The equations for measuring the lengths of the DIA and ZIA wiring harnesses designed in this paper can be expressed as follows. Equation (3) shows the definition of the total lengths of the wiring harnesses in DIA.
(3)$$L_{DIA}=\overset{N_{DC}-1}{\underset{i=1}{\sum }}L_{DC_{i}}+\overset{N_{DC}-1}{\underset{i=1}{\sum }}\overset{N_{DE}}{\underset{j=1}{\sum }}\alpha L_{DE_{\left(i,\;\;j\right)}}$$

In (3), $L_{DIA}$ is the total lengths of the wiring harnesses in DIA. $N_{DC}$ and $N_{DE}$ are the numbers of the domain controllers and the ECUs mounted in DIA. $L_{DC_{i}}$ is the lengths of the wiring harnesses between the $i^{th}$ domain controller and the gateway domain controller. $L_{DE_{\left(i,j\right)}}$ is the length of the wiring harnesses between the $i^{th}$ domain controller and the $j^{th}$ ECU. The term α is the variable value depending on the length of the wiring harnesses. Since DIA is a structure that connects ECUs and a domain controller according to function regardless of distance, the lengths of the wiring harnesses are different. The larger the α value, the farther away the ECU is from the domain controller.
(4)$$L_{ZIA}=\overset{N_{ZC}-1}{\underset{i=1}{\sum }}L_{ZC_{i}}+\overset{N_{ZC}-1}{\underset{i=1}{\sum }}\overset{N_{ZE}}{\underset{j=1}{\sum }}L_{ZE_{\left(i,j\right)}}$$

Since ZIA groups ECUs according to their physical location, it is assumed that the length of the wiring harnesses between the ECUs mounted inside the zone and the zone controller is the same. In (4), $L_{ZIA}$ is the total lengths of the wiring harnesses in ZIA. $N_{ZC}$ and $N_{ZE}$ are the numbers of the zone controllers and the ECUs mounted in ZIA. $L_{ZC_{i}}$ is the length of the wiring harnesses between the $i^{th}$ zone controller and the gateway zone controller. $L_{ZE_{\left(i,\;j\right)}}$ is the length of the wiring harness between the $i^{th}$ zone controller and the $j^{th}$ ECU.

The length of the wiring harnesses of DIA and ZIA proposed in this paper were estimated by referring to the size of a mid-size vehicle. The size of the referenced mid-size vehicle is assumed to be 4.8 m in overall length and 1.6 m in width considering a mid-size vehicle. Since measuring the lengths and weights of the wiring harnesses used in the actual vehicle was difficult, we referred to the wiring harness information of the Volkswagen Golf 7. According to [26], the wiring harness used in this vehicle is about 1.6 km in total lengths and weighs up to 60 kg. The estimated total weights of the wiring harnesses are 0.0375 kg per m. We design the IVN architectures with reference to this information and then calculated the lengths and weights of the wiring harnesses required to design the IVN architectures. The total weights of DIA and ZIA wiring harnesses can be calculated by multiplying $L_{DIA}$ and $L_{ZIA}$ by the weight per m.

## 5. Performance Evaluation for DIA and ZIA through Simulation

Section 5 describes the results obtained through the DIA and ZIA simulators developed in the previous section. Table 2 shows the average E2E delays measured in DIA and ZIA. The SR Class A traffic includes audio, HUD, and driver instrument display data. The SR Class B traffic includes entertainment data and GPS data and image and AVM, rear side, and mirrorless images. The ST includes control, navigation, LiDar, PT and chassis, and V2X data. Figure 7, Figure 8 and Figure 9 show comparison results of average E2E delays measured by DIA and ZIA according to data transmission priority.

Figure 7 compares E2E delays for the SR Class A and B traffic in DIA and ZIA through simulation. The x-axis represents the traffic types, and the y-axis represents the E2E delays. The E2E delay for the audio data is 89.43 μs in DIA and 84.93 μs  in ZIA. The E2E delay for the HUD data is 486.05 μs in DIA and 91.94 μs in ZIA. Finally, the E2E delays for the driver instrument display data is 486.05 μs in DIA and 91.94 μs in ZIA. We verified that the E2E delays for the SR Class A traffic be shorter in ZIA than in DIA through simulation. The E2E delay for the entertainment data is 365.75 μs in DIA and 365.08 μs in ZIA. The E2E delay for the GPS data and image is 572.87 μs in DIA and 159.84 μs in ZIA. The E2E delay for the AVM image is 573.05 μs in DIA and 160.2 μs in ZIA. The E2E delay for the rear side image is 573.05 μs on DIA and 160.34 μs on ZIA. Finally, the E2E delay for the mirrorless image is 573.09 μs in DIA and 160.22 μs in ZIA. We verified that the E2E delays for the SR Class B traffic be shorter in ZIA than in DIA through simulation.

Figure 8 compares the E2E delays for the ST with the highest priority in DIA and ZIA. The E2E delay for the control data (1) is 36.68 μs in DIA and 19.52 μs in ZIA. The E2E delay for the control data (2) is 92.13 μs  in DIA and 65.65 μs in ZIA. The E2E delay for the control data (3) is 92.15 μs in DIA and 78.9 μs in ZIA. The E2E delay for the PT and chassis data is 51.05 μs in DIA and 30.89 μs  in ZIA. The E2E delay for the V2X data is 31.51 μs in DIA and 14.34 μs in ZIA. However, the E2E delay for the navigation data is 9.16 μs in DIA and 17.74 μs in ZIA. Finally, the E2E delay for the LiDar data are the same for DIA and ZIA. We verified that the E2E delays for five out of seven data be shorter in ZIA than in DIA through simulation.

Figure 9 shows the results of comparing the lengths and weights of the wiring harnesses required to design DIA and ZIA proposed in this paper. The necessary wiring harnesses for DIA and ZIA are turned out to be 65.05 m and 49.02 m, respectively. Comparing the weights of the wiring harnesses for the two architectures, we found that the weights are 2.44 kg for DIA and 1.84 kg for ZIA, respectively. Unlike DIA, which connects ECUs with similar functions to one controller regardless of distance, ZIA connects to the closest controller based on the physical location of ECUs. For this reason, when ZIA is adopted for IVNs, the lengths and weights of the wiring harnesses can be reduced compared to DIA.

## 6. Conclusions and Future Works

Recently, OEMs and vehicle component manufacturers looking to reduce weight and cost and simplify connectivity in their IVN designs have begun turning to ZEA. Because ZIA groups ECUs according to their physical location, it has the advantage of reducing the lengths and weights of wiring harnesses compared to previous architectures. In this paper, ZIA and DIA were compared to verify the advantages of ZIA. For comparison, we designed new DIA and ZIA for autonomous vehicles and developed simulators to evaluate the performance of these architectures. The performance was compared and analyzed in terms of E2E delay and the total length and weight of the wiring harnesses using a simulator.

First of all, using the DIA and ZIA simulators, we verified that ZIA could reduce the E2E delays by approximately 36.7% compared to DIA. The E2E delays are lower in ZIA than in DIA because data are transferred over fewer switches and links in ZIA than in DIA. We developed a simulator by selecting ECUs closely related to autonomous driving. Not limiting the number of ECUs not only increases the amount of data that needs to be processed inside the links and switches but also increases the processing time and delays. Although the number of ECUs is limited to 36 in this paper, if more than 100 ECUs are simulated, the difference in E2E delays between DIA and ZIA will become even more significant. Therefore, ZIA is judged to be effective as an IVN architecture for autonomous driving or future vehicles, where ECUs are expected to increase.

In addition, we verified that ZIA could reduce the lengths and weights of the wiring harness by 24.6% compared to DIA. As the number of ECUs increases, the lengths of wiring harnesses required for the connections between ECUs would inevitably increase. According to [15], the total length of the wiring harnesses for a vehicle with about 150 ECUs is 5 km. Autonomous vehicles and future vehicles would have more ECUs than vehicles today. The total length of the wiring harnesses would be increased when DIA, which is a method of grouping ECUs by functions and connecting the controller and ECUs regardless of distance, is adopted in the vehicle. Therefore, adopting ZIA in vehicles may be an inevitable option but a necessity to develop autonomous vehicles and future vehicles.

The ZIA proposed in this paper adopted a star topology. In future research, topologies such as ring, tree, and mesh may be adopted to ZIA. In addition, future research remains on finding the correlation between the number of ECUs mounted on a vehicle and the lengths and weights of the wiring harnesses.

## Figures and Tables

**Figure 1 sensors-23-00669-f001:**
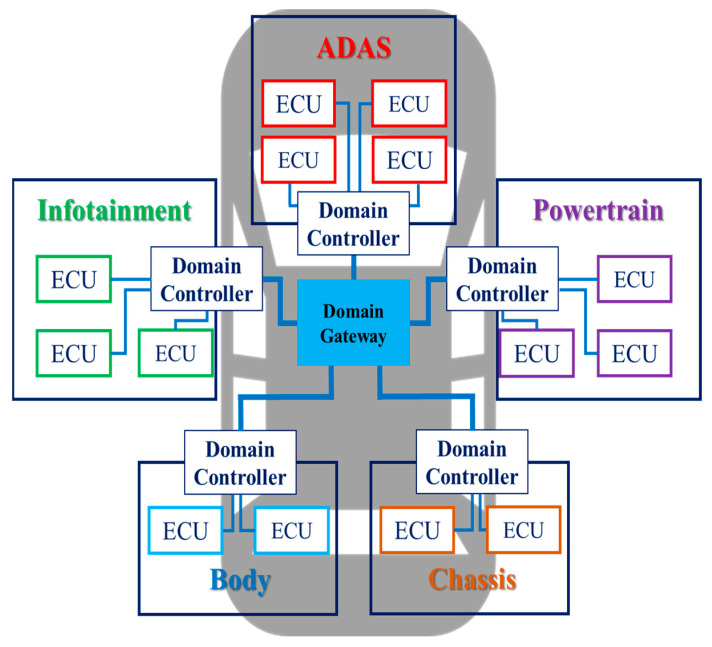
DIA with multiple functional domains interconnected by a domain gateway.

**Figure 2 sensors-23-00669-f002:**
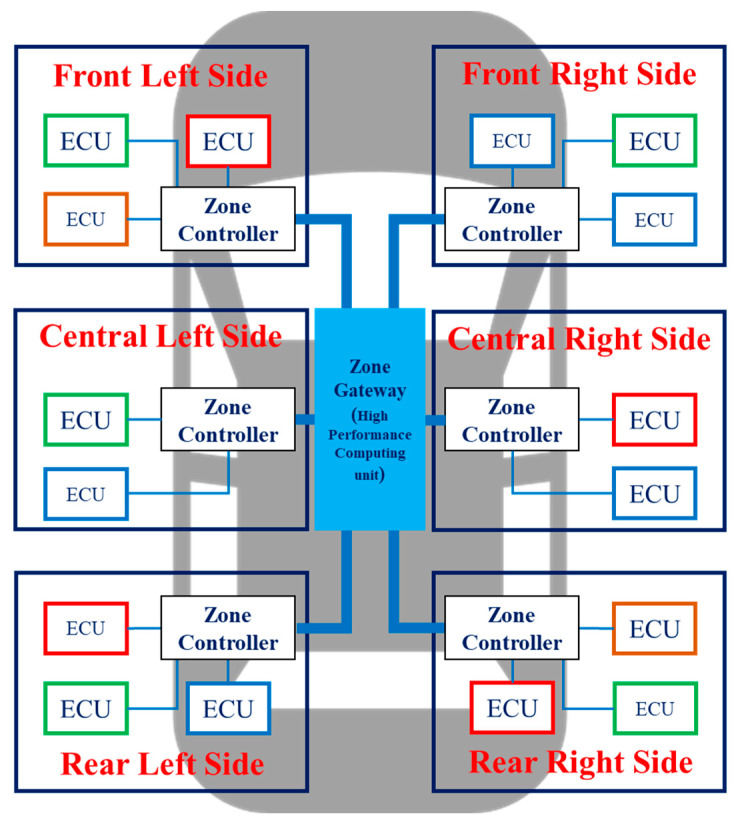
ZIA with a zone gateway and multiple zone controllers.

**Figure 3 sensors-23-00669-f003:**
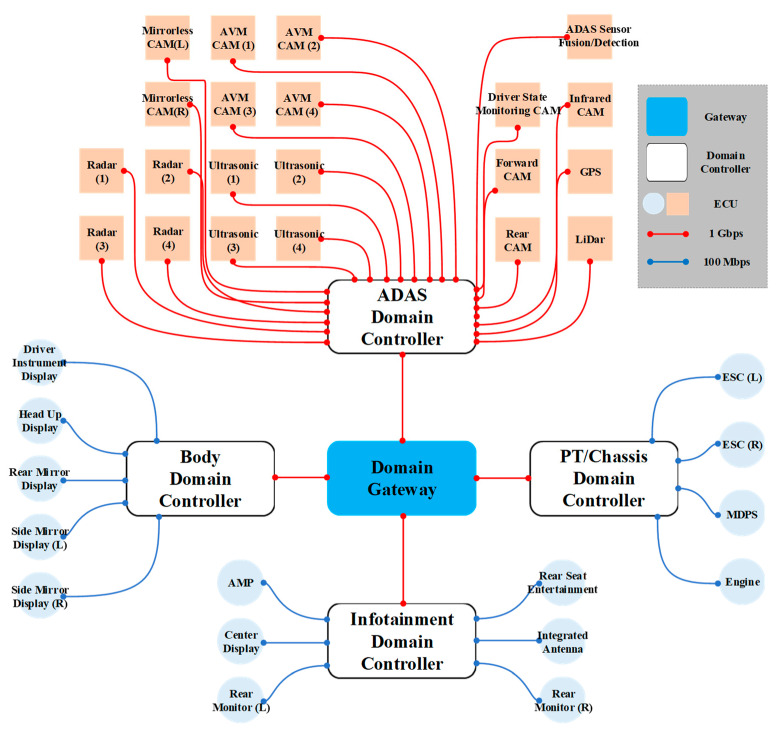
Design of DIA grouped into four domains with automotive Ethernet.

**Figure 4 sensors-23-00669-f004:**
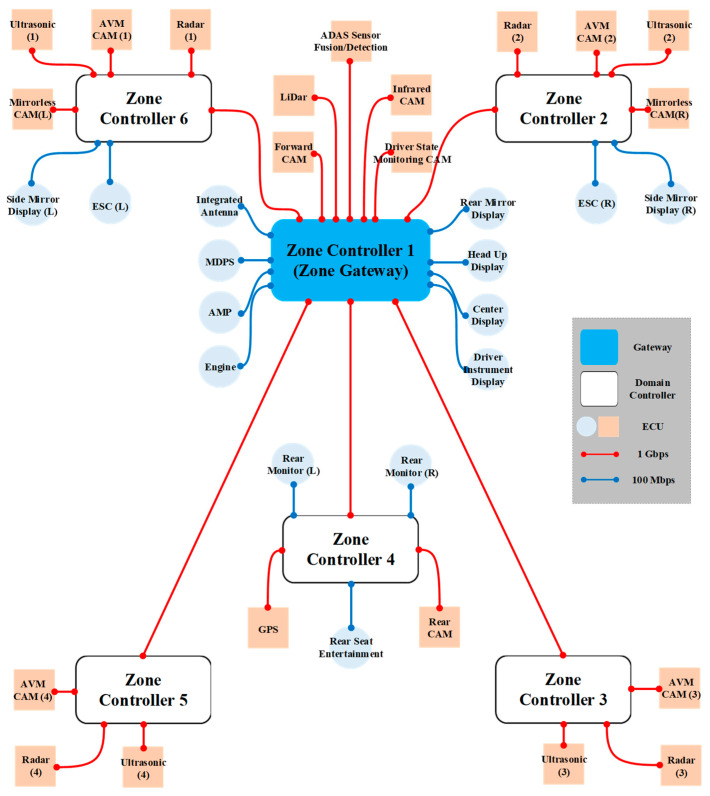
Design of ZIA grouped into six zones with automotive Ethernet.

**Figure 5 sensors-23-00669-f005:**
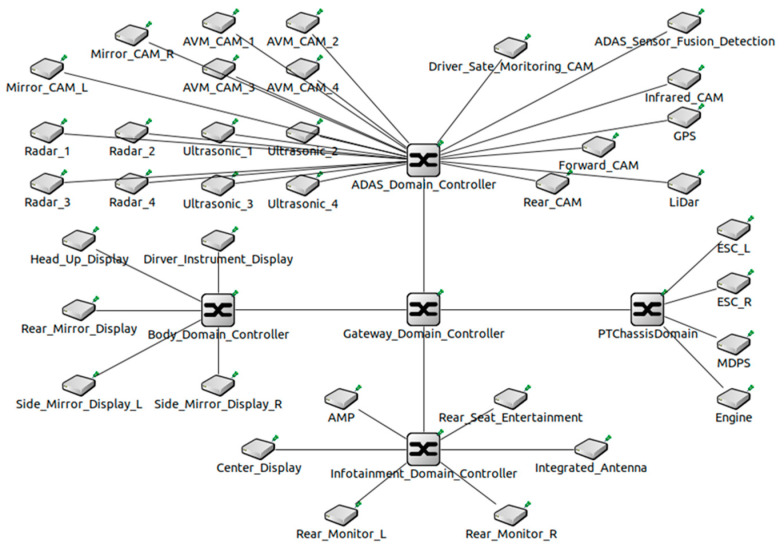
Development of DIA grouped into four domains such as ADAS, PT and chassis, infotainment, and body using the OMNeT++.

**Figure 6 sensors-23-00669-f006:**
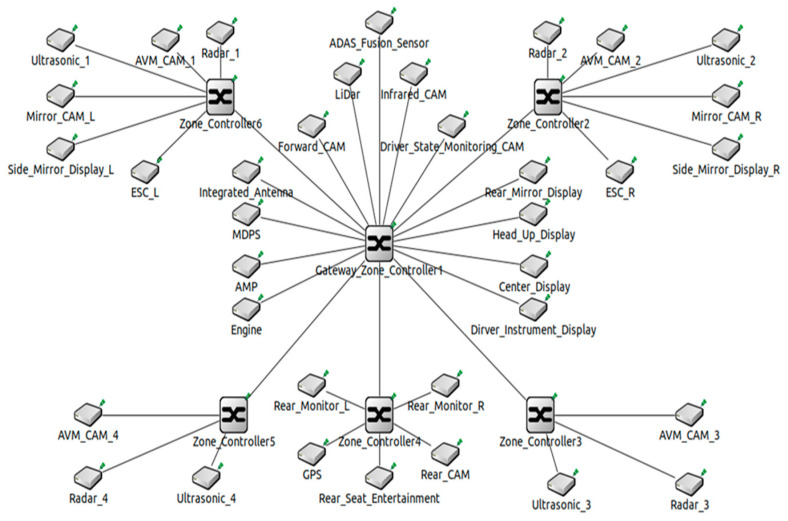
Development of ZIA grouped into six zones using the OMNeT++.

**Figure 7 sensors-23-00669-f007:**
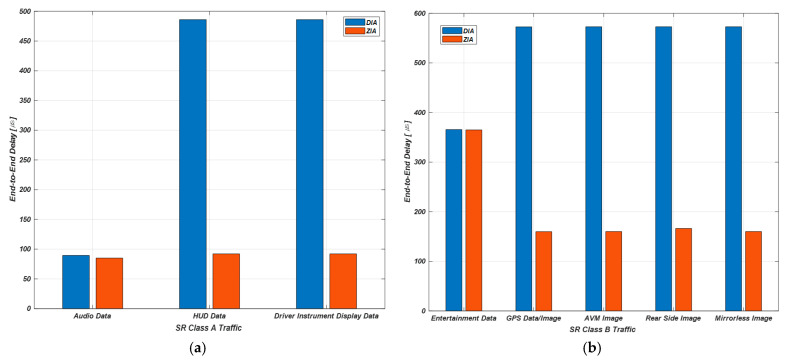
Comparison of E2E delays for the SR Class A and B. (**a**) E2E delays for the SR Class A; (**b**) E2E delays for the SR Class B.

**Figure 8 sensors-23-00669-f008:**
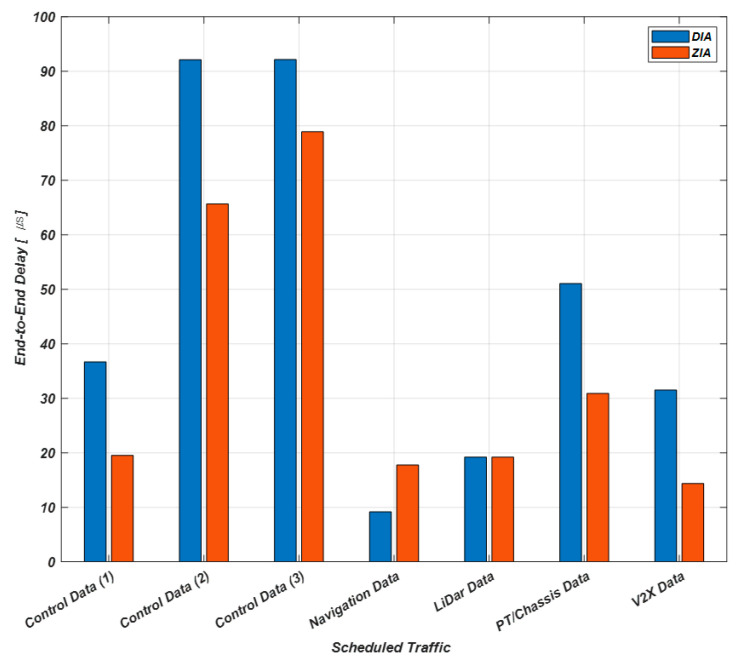
Comparison of E2E delays for the ST with the highest priority.

**Figure 9 sensors-23-00669-f009:**
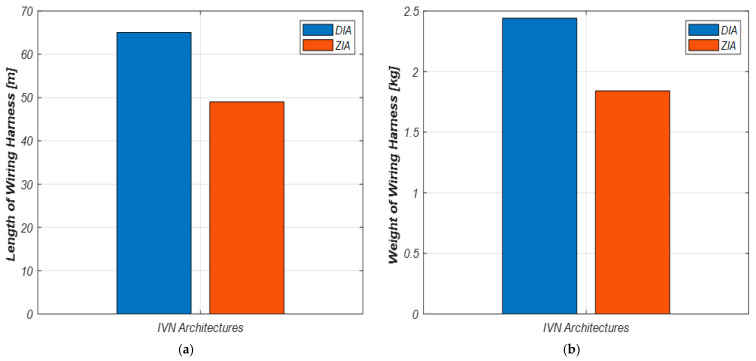
Comparison of the total length and weight of the wiring harness for the two IVN architectures designed in this paper. (**a**) Comparison of the total lengths of the wiring harnesses for DIA and ZIA; (**b**) Comparison of the total weights of the wiring harnesses for DIA and ZIA.

**Table 1 sensors-23-00669-t001:** Information on traffic generated by ECUs in DIA and ZIA.

Traffic Type	Traffic Priority	Payload Size [Byte]	Min. Traffic Interval [μs]	Transmission Path
Audio data	SR Class A	11	125	Rear Seat Entertainment → AMP
Entertainment data	SR Class B	1250	250	Rear Seat Entertainment → Rear Left/Right Monitor
Control data (1)	ST	4	500	ADAS Sensor FD → Engine
AVM image	SR Class B	1250	250	AVM CAM → Center Display
Rear side image	SR Class B	1250	250	Rear CAM → Center Display
Navigation data	ST	12	500	GPS → ADAS Sensor FD
LiDar data	ST	670	500	LiDar → ADAS Sensor FD
PT/Chassis data	ST	230	500	MDPS → ADAS Sensor FD
V2X data	ST	16	500	Integrated Antenna → ADAS Sensor FD
Control data (2)	ST	625	500	ADAS Sensor FD → MDPS
Control data (3)	ST	625	500	ADAS Sensor FD → ESC
GPS data/image	SR Class B	1250	250	ADAS Sensor FD → Center Display
Mirrorless image	SR Class B	1250	250	ADAS Sensor FD → Side Mirror Display
HUD data	SR Class A	38	125	ADAS Sensor FD → HUD
Driver instrument display data	SR Class A	10	125	ADAS Sensor FD → Driver Instrument Display

**Table 2 sensors-23-00669-t002:** Comparison of average end-to-end delays for the traffic types according to IVN architectures.

Traffic Priority		Traffic Type	Average of End-to-End Delay [μs]	
			DIA	ZIA
AVB	SR Class A	Audio Data	89.43	84.93
		HUD Data	486.05	91.94
		Driver Instrument Display Data	486.05	91.94
	SR Class B	Entertainment Data	365.75	365.08
		GPS Data/Image	572.87	159.84
		AVM Image	573.05	160.2
		Rear Side Image	573.05	160.34
		Mirrorless Image	573.09	160.22
ST		Control Data (1)	36.68	19.52
		Control Data (2)	92.13	65.65
		Control Data (3)	92.15	78.9
		Navigation Data	9.16	17.74
		LiDar Data	19.21	19.21
		PT/Chassis Data	51.05	30.89
		V2X Data	31.51	14.34

## Data Availability

Not applicable.

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
