# Peer review of "Performance Evaluation of Zone-Based In-Vehicle Network Architecture for Autonomous Vehicles"

_sensors, 2023, doi:10.3390/s23020669_

Round 1

Reviewer 1 Report

The authors presented a Performance Evaluation of Zone-based In-Vehicle Network Architecture. The paper is well presented. 

Please clearly mention the contributions in the introduction section.

Compared to the listed paper in the related work section, please add the advantages of the proposed method in the conclusion section or may add a section before conclusion named DIscussion. 

Author Response

Dear editor and reviewer 1,

Thanks very much for taking the time to review this manuscript. We thank the reviewers for the time and effort they put into reviewing the previous version of the manuscript. Their suggestions have enabled us to improve our work.

Point 1:

Please clearly mention the contributions in the introduction section. Compared to the listed paper in the related work section, please add the advantages of the proposed method in the conclusion section or may add a section before conclusion named Discussion.

Response 1:

We corrected the introduction and conclusion sections. Modified parts are marked in red. Please check the attached file.

Best regards,

Reviewer 2 Report

In order to minimize the complexity of wiring harness and IVN, this study proposes a new DIA and ZIA architecture in the context of time-sensitive networks. It's interesting, but we still have the following concerns:

(1) In the abstract, this paper proposes to design a new DIA and ZIA architecture, but in Chapter 3, it does not highlight how is it different from the previous structure, and what are the advantages over the previous structure, please explain in detail.

(2) The title of this paper is to evaluate the performance of ZIA, but the study mainly compares with DIA to highlight the performance advantages of ZIA. Is it representative? On one hand, is the selection of evaluation indicators reasonable and comprehensive? On the other hand, is it persuasive to compare with DIA only in terms of performance?

(3) In order to make the article more substantial and comprehensive, we can make a vertical comparison in the context of time-sensitive networks, and analyze the performance of ZIA architecture changes with different scenario parameters by using line charts and other methods to demonstrate its practical value.

Author Response

Dear editor and reviewer 2,

Thanks very much for taking the time to review this manuscript. We thank the reviewers for the time and effort they put into reviewing the previous version of the manuscript. Their suggestions have enabled us to improve our work.

Point 1:

In the abstract, this paper proposes to design a new DIA and ZIA architecture, but in Chapter 3, it does not highlight how is it different from the previous structure, and what are the advantages over the previous structure, please explain in detail.

Response 1:

Prior to DIA and ZIA, a distributed in-vehicle network (IVN) architecture was used. In subsection 2.2, we explained the advantages of DIA and ZIA over distributed IVN architectures. The new DIA and ZIA mentioned in the abstract are architectures designed by selecting 36 ECUs related to autonomous driving for simulation.

The introduction and section 3 have been revised. Modified parts are marked in red. Please check the attached file. Thank you.

Point 2:

The title of this paper is to evaluate the performance of ZIA, but the study mainly compares with DIA to highlight the performance advantages of ZIA. Is it representative? On one hand, is the selection of evaluation indicators reasonable and comprehensive? On the other hand, is it persuasive to compare with DIA only in terms of performance?

Response 2:

DIA is an architecture that is currently being applied to vehicles. In this regard, this paper compared ZIA and DIA, which have recently attracted attention. There are various elements for performance evaluation (Throughput, Speed, Latency, Delay, Error, Cost, etc.). This paper focused on three evaluation elements. These reasons include the following.

1) In time-sensitive networks, the most important performance measure is E2E delay. The TSN standards specify the amount of time required to transmit traffic.

2) The main advantage of ZIA is that it reduces the lengths and weights of the wiring harnesses compared to DIA. Length and weight reduction are mentioned in many technical documents and the like. However, information on how much can be reduced numerically is lacking.

Therefore, the simulation focused on E2E delay and the total length and weight of the wiring harnesses. Thank you.

Point 3:

In order to make the article more substantial and comprehensive, we can make a vertical comparison in the context of time-sensitive networks, and analyze the performance of ZIA architecture changes with different scenario parameters by using line charts and other methods to demonstrate its practical value.

Response 3:

In reflection of the reviewer's opinion, I tried to print the graph in various ways other than the line chart. However, the vertical graph used in this paper is judged to be easier to understand than other graphs. When I write another paper in the future, I will use various graphs to express the results. Thank you.

Best regards,

Round 2

Reviewer 2 Report

The paper is appropriate for publication. I have no more questions.